# FpnA, the *Aspergillus fumigatus* homolog of human ferroportin, mediates resistance to nickel, cobalt and gallium but does not function in iron homeostasis

Isidor Happacher [1,3], Simon Oberegger [1,3], Beate Abt[1], Annie Yap[1], Patricia Caballero [1], Mario Aguiar [1], Javeria Pervaiz[1], Giacomo Gariglio[2], Matthias Misslinger [1], Clemens Decristoforo [2] & Hubertus Haas [1] ✉

Iron homeostasis is key to both the survival of virtually all organisms and the virulence of fungi including *Aspergillus fumigatus*, a human fungal pathogen causing life-threatening invasive infections. Unlike the extensively studied fungal species *Saccharomyces cerevisiae* and *Schizosaccharomyces pombe*, *A. fumigatus* encodes an uncharacterized homolog of vertebrate ferroportin (Fpn1), termed FpnA. Fpn1 is the only known vertebrate iron efflux transporter, while microbial organisms are thought to lack iron efflux systems. After correcting the exon-intron annotation, inactivation and conditional overexpression of the *A. fumigatus* FpnA-encoding gene (*fpnA*) indicated, that FpnA mediates resistance to nickel, cobalt and gallium but not to iron, aluminium, cadmium, copper or zinc. Functional N-terminal tagging with a fluorescent protein demonstrated localization of FpnA in the vacuolar membrane, suggesting that FpnA detoxifies substrate metals by vacuolar deposition. In line, overexpression of *fpnA* reduced the utilization of urea as a nitrogen source, most likely by depriving cytosolic urease of its essential cofactor nickel. Phylogenetic analysis illustrated conservation of FpnA in all fungal divisions and several other eukaryotic lineages, underlining its crucial role in metal homeostasis. The divergent localization and functionalization of ferroportin homologs in two phylogenetic sister groups, metazoa and fungi, is of particular evolutionary interest.

Early in evolution, iron was employed by various biochemical pathways and now plays a critical role in the functioning of virtually all life forms[1]. Although iron is an essential element, it is highly toxic in excess causing oxidative stress[2]. As a result, iron homeostasis must be tightly controlled. In recent years, *Aspergillus fumigatus* became a role model for iron homeostasis and its role in virulence of filamentous fungi[3]. *A. fumigatus* is a ubiquitous saprobic mold growing on decaying organic material but, at the same time, it represents the most common human mold pathogen causing life-threatening invasive infections in immunocompromised patients[4]. This mold employs two high-affinity iron acquisition systems: siderophore-mediated iron uptake, which is essential for virulence in invertebrate and mammalian infection models[5–10], and reductive iron assimilation[10–12]. Iron detoxification is mediated by vacuolar iron deposition using the vacuolar

membrane transporter CccA[13]. Unlike vertebrates, microbial organisms including fungal species are assumed to lack cellular iron efflux systems.

Maintenance of iron homeostasis in *A. fumigatus* involves two transcription factors, termed as SreA and HapX, which sense the cellular iron status by binding iron-sulfur clusters[3,14–16]. SreA counteracts excessive iron uptake by repressing high-affinity iron acquisition; HapX mediates adaptation to both iron starvation and iron excess by controlling iron-consuming pathways such as the TCA cycle, respiration and heme biosynthesis as well as vacuolar iron deposition[12].

Unlike the intensively studied fungal species *Saccharomyces cerevisiae*, *Candida albicans* and *Schizosaccharomyces pombe*, the *A. fumigatus* genome encodes an uncharacterized homolog of the human iron transporter ferroportin (Fpn1), termed FpnA (Afu5g12920). While many molecular

[1]Institute of Molecular Biology, Biocenter, Medical University Innsbruck, Innsbruck, Austria. [2]Department of Nuclear Medicine, Medical University Innsbruck, Innsbruck, Austria. [3]These authors contributed equally: Isidor Happacher, Simon Oberegger. ✉e-mail: hubertus.haas@i-med.ac.at

iron handling systems have redundancies, the membrane protein ferro-portin is the only known vertebrate iron efflux system responsible for transporting intestinally absorbed, or stored, iron into the blood for circulation[17,18]. In contrast, microbial organisms including fungal species are assumed to lack cellular iron efflux systems.

In the current study we corrected the exon-intron annotation of the *A. fumigatus* FpnA-encoding gene and characterized its function and phylogenetic distribution.

## Results

### The *A. fumigatus* genome encodes a ferroportin homologue

Homology searches using the Basic Local Alignment Search Tool (blastp; https://blast.ncbi.nlm.nih.gov/Blast.cgi) revealed that, unlike the intensively studied fungal species *Saccharomyces cerevisiae*, *Candida albicans* and *Schizosaccharomyces pombe*, *A. fumigatus* possesses a homolog (Afu5g12920) to human Fpn1, termed FpnA. However, the predicted exon-intron structure and deduced protein sequence of the *A. fumigatus fpnA*-encoding gene deposited in NCBI (Accession number: XP_753361.1) or FungiDB (Accession number: Afu5g12920) did not match the transcriptomics data deposited in FungiDB. The manually corrected exon-intron structure and protein sequence, Afu5g12920-T, is deposited under *Community annotations from Apollo* in FungiDB (Supplementary Fig. S1). The corrected *A. fumigatus* FpnA and human Fpn1 proteins show an identity of 25% and are the best homologs in reverse blastp searches with an E-value of 3e[-47] for FpnA versus the human proteome.

### FpnA mediates resistance to nickel, cobalt and gallium

To functionally characterize FpnA, the encoding *fpnA* gene was deleted in *A. fumigatus* AfS77, termed wild type (WT) here, by replacement with the hygromycin resistance marker gene (*hph*), resulting in the strain Δ*fpnA*. Moreover, *fpnA* was conditionally expressed under control of the *xylP* promoter (P*xylP*) with (*fpnA*^P*xylP-Venus* strain) or without (*fpnA*^P*xylP* strain) N-terminal tagging with the yellow fluorescence protein derivative Venus[19–21]. P*xylP* allows high induction in the presence of xylose and shows low basal expression in the absence of xylose[19,20].

For phenotyping, the fungal strains were grown on solid minimal media without and with 1% xylose for strong induction of the *xylP* promoter[19,20]. Neither deletion (Δ*fpnA* strain) nor downregulation (*fpnA*^P*xylP* and *fpnA*^P*xylP-Venus* strains without xylose) or overexpression (*fpnA*^P*xylP* and *fpnA*^P*xylP-Venus* strains with xylose) of *fpnA* affected the growth pattern on standard minimal medium containing 0.01 mM iron (Fig. 1A). Due to the iron export function of human Fpn1, we expected a function of FpnA in iron resistance. However, all generated mutant strains showed WT-like growth on media containing 6 mM or 10 mM iron (Fig. 1A). Even in the absence of the iron-regulatory transcription factor SreA, which decreases iron tolerance due to derepression of iron uptake[22], deletion (Δ*fpnA*Δ*sreA* strain with and without xylose) nor downregulation (*fpnA*^P*xylP* strain without xylose) or overexpression (*fpnA*^P*xylP*Δ*sreA* strain with xylose) affected growth compared to Δ*sreA* on media containing 0.01 mM or 6 mM iron (Fig. 1B). Notably, the toxicity of iron excess is reflected by the decreased radial growth of SreA-possessing strains on 10 mM iron (Fig. 1A) and SreA-lacking strains on standard medium and in the presence of 6 mM iron (Fig. 1B).

Functional studies in *Xenopus* oocytes demonstrated that human Fpn1 transports metals other than iron, such as cobalt, zinc, and nickel, but not copper or cadmium[23,24]. Therefore, we tested the potential function of *A. fumigatus* FpnA in resistance to other metals (Fig. 1A). Deletion of *fpnA* (Δ*fpnA* strain) blocked growth in the presence of high concentration of nickel, and decreased growth on cobalt and gallium. In line, downregulation of *fpnA* (*fpnA*^P*xylP* and *fpnA*^P*xylP-Venus* strains without xylose) tended to decrease resistance to these metals. Interestingly, strain *fpnA*^P*xylP* showed higher resistance compared to strain *fpnA*^P*xylP-Venus*, which indicates that Venus tagging partially impairs FpnA function. Strains with xylose-induced *fpnA* expression (*fpnA*^P*xylP* and *fpnA*^P*xylP-Venus* strains with xylose) displayed significantly higher resistance to nickel, cobalt and gallium than WT. Notably, *fpnA*^P*xylP* showed higher resistance compared to *fpnA*^P*xylP-Venus*,

which again indicates that Venus-tagging slightly impairs FpnA function. Nevertheless, these data underline that the Venus-tagged FpnA version is functional. In contrast, neither deletion nor downregulation or overexpression of *fpnA* affected growth on high concentrations of copper, cadmium, aluminum and zinc (Fig. 1A).

Taken together, these results demonstrate that FpnA is involved in detoxifying nickel, cobalt and gallium but not iron, copper, cadmium, aluminum and zinc.

### The *fpnA* transcript levels are not affected by exposure to the metal substrates

In Northern blot analysis, the *fpnA* transcript level was not affected by a two-hour confrontation with high concentrations of nickel, copper, cobalt, gallium or iron indicating constitutive transcription of this gene (Fig. 2A). In contrary, *cccA* was induced by iron excess, as shown previously[13]. Moreover, Northern blot analysis confirmed xylose-induced *fpnA* expression in *fpnA*^P*xylP* and *fpnA*^P*xylP-Venus* (Fig. 2B), which is in agreement with the xylose-induced metal resistance of these strains (Fig. 1A). Furthermore, Northern blot analysis revealed significantly higher *fpnA* transcript levels in xylose-induced *fpnA*^P*xylP* and *fpnA*^P*xylP-Venus* compared to WT (Fig. 2C), which suggests that the higher metal resistance of *fpnA*^P*xylP* and *fpnA*^P*xylP-Venus* compared to WT (Fig. 1A) is due to higher *fpnA* expression.

Taken together, these data indicate constitutive expression of *fpnA* in *A. fumigatus* and demonstrate that FpnA-mediated metal resistance can be increased by elevated *fpnA* expression.

### FpnA is localized in the vacuolar membrane

Epifluorescence microscopy (Fig. 3) revealed that Venus-tagged FpnA localizes to the vacuolar membrane, with the vacuolar lumen visualized using CellTracker™ Blue CMAC Dye[25]. Based on the same transport direction as human Fpn1[26], these results suggest that FpnA most likely detoxifies the substrate metals by transport from the cytoplasm into the vacuolar lumen.

### Overexpression of *fpnA* decreases utilization of urea as sole nitrogen source

Unlike to cobalt and gallium, nickel is essential for a biological process as it is a co-factor for the urea-degrading enzyme urease[27]. In contrast to WT and Δ*fpnA*, *fpnA*^P*xylP* and *fpnA*^P*xylP-Venus* strains displayed reduced growth on urea as sole nitrogen source in the presence but not in the absence of xylose (Fig. 4). The reduced utilization of urea as a nitrogen source caused by the overproduction of FpnA can most likely be explained by the deprivation of the essential nickel co-factor of cytosolic urease through vacuolar nickel deposition. In agreement, supplementation with 0.5 mM nickel rescued growth of *fpnA*^P*xylP* and *fpnA*^P*xylP-Venus* in the presence of xylose, while this concentration largely blocked growth of WT and Δ*fpnA* as well as of *fpnA*^P*xylP* and *fpnA*^P*xylP-Venus* in the absence of xylose. The fact that overexpression of *fpnA* (*fpnA*^P*xylP* and *fpnA*^P*xylP-Venus* strains with xylose) did not cause growth defects on standard minimal medium with glutamine as nitrogen source (Fig. 1), suggests that FpnA does not transport metals, other than nickel, that are essential for cellular functions.

To further investigate the growth defect on urea caused by *fpnA* overexpression, we analyzed the expression of the *A. fumigatus* urease encoding gene, termed *ureB*[27], by Northern blot analysis. Interestingly, nickel supplementation and *fpnA* deletion, but not *fpnA* overexpression, decreased *ureB* transcript levels (Supplementary Fig. S2), possibly indicating negative regulation of *ureB* expression by nickel. The link between *ureB* transcript levels and nickel could be indirect through the positive effect of nickel on urease activity (see below), i.e., increased urease activity increases urea utilization, which could downregulate *ureB* expression. Urease activity, i.e., hydrolysis of urea, leads to formation of ammonia and hence alkalinization, which can be detected by the pH-sensitive color change of phenol red[27]. Nickel supplementation enhanced the reddish coloration of mycelia of WT, Δ*fpnA* and *fpnA*^P*xylP* strains (Supplementary Fig. S3), indicating increased urease activity. These results suggest that nickel is a limiting factor

**Fig. 1 | FpnA mediates resistance to nickel, cobalt and gallium but not to iron, cadmium, aluminum or zinc.** Fungal strains possessing (**A**) or lacking SreA (**B**) were spot-inoculated on solid minimal medium without (-) or with the indicated metal supplementation without or with 1% xylose for induction of the P*xylP* promoter. Plates were incubated at 37 °C for 48 h.

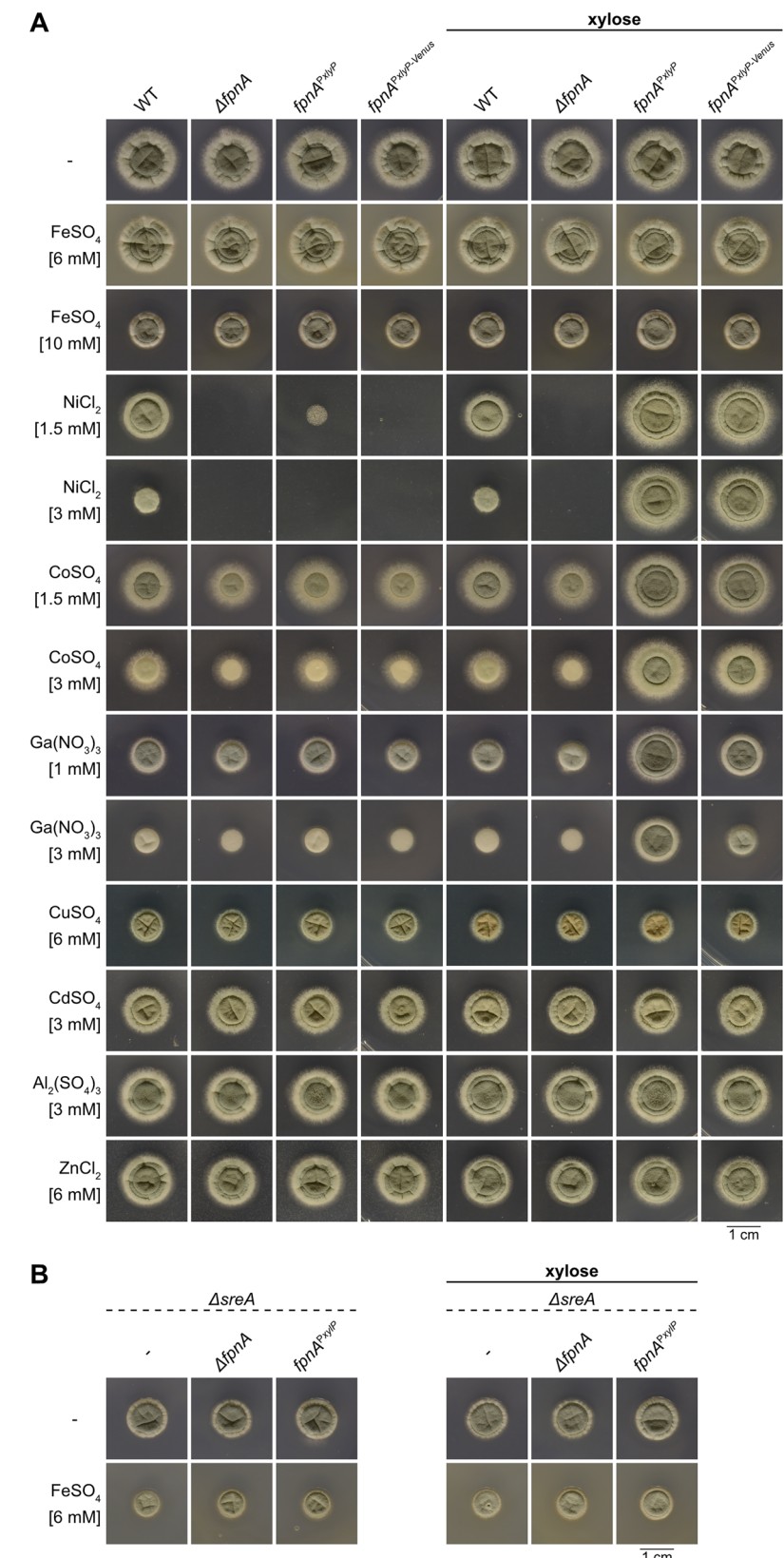

for urease activity in the growth medium used. Compared to WT and Δ*fpnA* strains, the *fpnA*^P*xylP* mutant displayed reduced reddish coloration of its mycelia without, but not with, nickel supplementation, supporting the hypothesis that *fpnA* overexpression reduces urease activity without nickel supplementation, most likely due to vacuolar nickel deposition.

## FpnA is highly conserved in the fungal kingdom

*A. fumigatus* FpnA is the first functional characterized fungal ferroportin homolog. Despite the fact that the intensively studied fungal species *S. cerevisiae C. albicans* and *S. pombe* lack FpnA homologs, FpnA was found to be conserved not only in Ascomycetes (in the classes Eurotiomycetes,

**Fig. 2 | Expression of *fpnA* is, in contrast to *cccA*, not affected by confrontation with the metal substrates. A** P*xylP*-control allows conditional expression of *fpnA* (**B**), and P*xylP*-control leads to higher expression of *fpnA* compared to its native promoter (**C**). For Northern analysis, fungal strains were grown for 18 h at 37 °C in liquid minimal medium flask cultures (-) or for 16 h followed by another 2 h incubation after addition of the indicated metal (**A**) or 1% xylose (**B**, **C**). NiCl$_2$, CoSO$_4$ and Ga(NO$_3$)$_3$ were added to a final concentration of 1.5 mM, CuSO$_4$ and FeSO$_4$ were added to a final concentration of 3 mM. The different concentrations were used due to the different toxicity of the metals (Fig. 1). In *fpnA*$^{PxylP-Venus}$, the *fpnA* transcript is larger due to fusion with Venus (**B**). For comparison of *fpnA* expression mediated by the P*xylP* versus the native promoter, fungal strains were grown for 18 hours at 37 °C in the presence of 1% xylose (**C**).

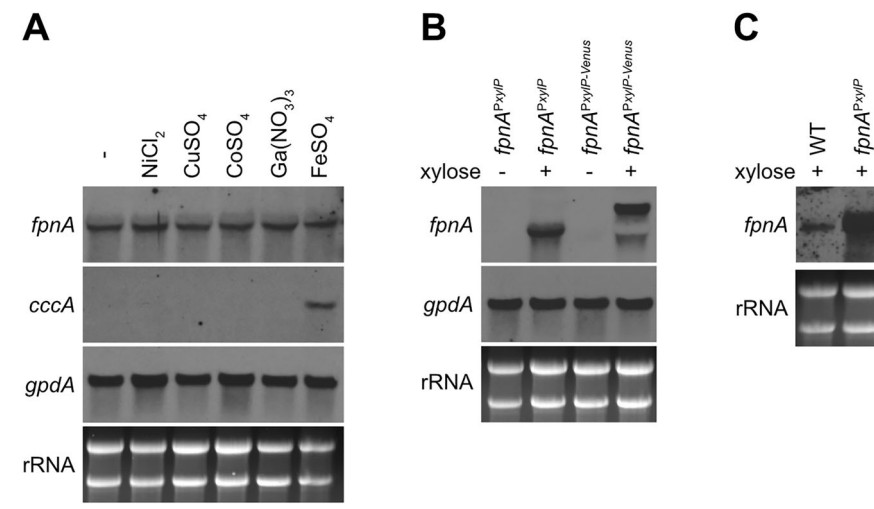

**Fig. 3 | FpnA is localized in the vacuolar membrane.** The Venus fluorescence of Venus-tagged FpnA (*fpnA*$^{PxylP-Venus}$ strain) was visualized using confocal microscopy. The vacuolar lumen was labelled using CellTracker™ Blue CMAC Dye.

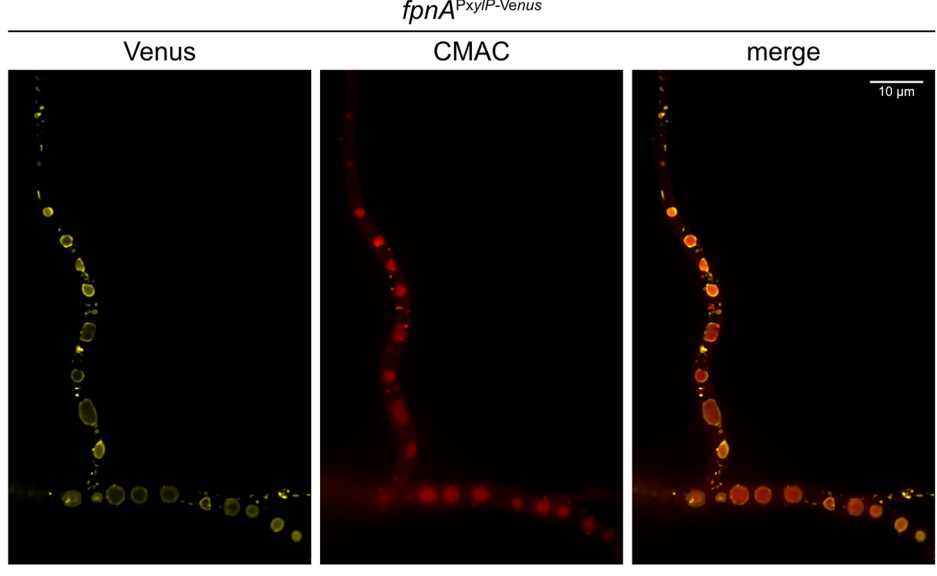

**Fig. 4 | Overexpression of *fpnA* (*fpnA*$^{PxylP}$ and *fpnA*$^{PxylP-Venus}$ strains with xylose) impairs utilization of urea as sole nitrogen source.** Fungal strains were spot-inoculated on solid minimal medium containing 20 mM urea as sole nitrogen source with or without supplementation with 0.5 mM NiCl$_2$ as well as 1% xylose for induction of the *xylP* promoter. Plates were incubated at 37 °C for 48 h.

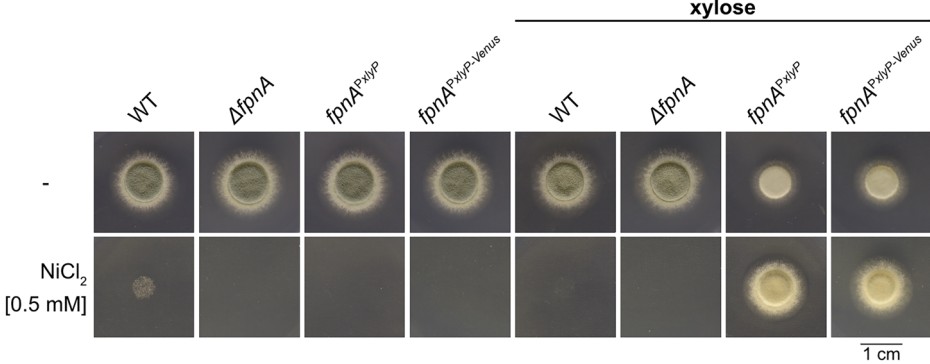

Dothideomycetes, Lecanoromycetes, Leotiomycetes, Orbiliomycetes, Pezizomycetes, Sordariomycetes, Xylonomycetes, and Saccharomycetes) but also in Basidiomycota (in the classes Agaricomycetes, Dacrymycetes, Tremellomycetes, Microbotryomycetes, Exobasidiomycetes, Ustilaginomycetes), Mucoromycota, Chytridiomycota and Blastocladomycota (Fig. 5 and

Supplementary Table S1), indicating an important role in diverse habitats. Fpn1 homologs from plants, termed IREG, have been reported to mediate cellular export as well as vacuolar storage of nickel in nickel-hyperaccumulating plant species[28,29]. Moreover, two Fpn1 homologs of *Arabidopsis thaliana*, localized in the plasma membrane and the vacuolar

**Fig. 5 | FpnA is highly conserved within the fungal kingdom, several other eukaryotic lineages and bacteria.** The phylogenetic analysis including selected species from different lineages was generated using Geneious Prime. The blast hits were aligned using the Clustal Omega method and the phylogenetic tree was constructed using the Neighbor-Joining method. The branches of the phylogenetic tree are represented as a cladogram. The kingdoms or phyla are colored in different shades of gray for better differentiation. Details of the displayed FpnA homologs are given in Supplementary Table S1.

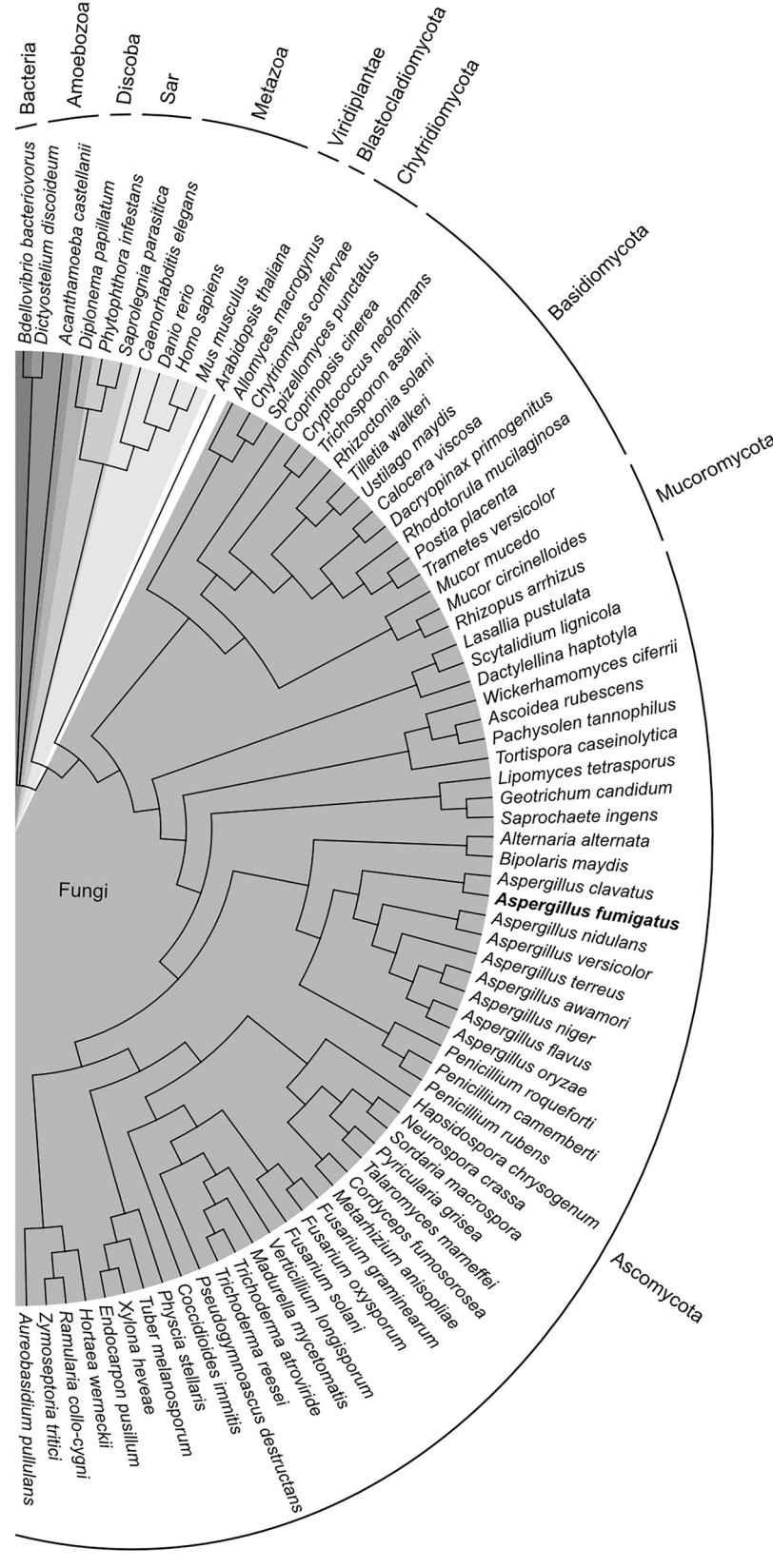

membrane, respectively, have been shown to function in iron and cobalt homeostasis[30]. As revealed by blastp searches (Fig. 5 and Supplementary Table S1), ferroportin homologs are found within Eukaryota not only in Metazoa (including fungi, mammals, Actinopterygii/fish and Nematoda) and Archaeplastida (including Viridiplantae/plants, green algae, red algae) but also in SAR (including the Stramenopiles/Oomycota species *Phytophthora infestans*), Discoba/Euglenozoa and Amoebozoa. In bacteria, ferroportin appears to be conserved only within the Gram-negative Bdellovibrionota phylum including free swimming *Bdellovibrio bacteriovorus*, which can prey on other Gram-negative bacteria. The physiological

substrate of the *B. bacteriovorus* Fpn1 homolog is still unknown. However, reconstituted into liposomes, it was shown to mediate in vitro transport of iron, cobalt, nickel and manganese and its crystal structure was elucidated[31].

Human Fpn1 displays a major facilitator superfamily (MFS)-type fold, comprising 12 transmembrane helices (TM) organized into N-terminal (TM1 - 6) and C-terminal (TM7 - 12) domains[31], with both N- and C-termini predicted to be cytosolic[26]. As shown in Supplementary Fig. S4, a ColabFold prediction (AlphaFold2 using MMseqs2) of *A. fumigatus* FpnA indicates a highly similar structure with a DALI Z-score of 26.3 for the human Fpn1 electron microscopy (EM) structure (PDB DOI: 6WBV, https://doi.org/10.2210/pdb6wbv/pdb)[32–34] and a DALI Z-score of 28.4 for the Fpn1 AlphaFold prediction (AF-Q9NP59-F1)[33,35]. A Z-score >20 indicates that the two structures are definitely homologous[33]. The AlphaFold structure prediction for human Fpn1 was included because several sequence parts were not modeled in the EM structure[32]: (i) the N-terminal region of Fpn1 (¹MTRAGDHNRQRGCCG[15]), (ii) the loop between TM6 and TM7 (²³⁹LKEEETELKQLNLHKDTEPKPLEGTHLMGVKDSNIHELEHEQEPTCASQM²⁸⁸), (iii) the loop between TM9 and TM10 (³⁹⁹DLSVSPFEDIRSRFIQGESITPTKIPEITTEIYMSNGSNSANIVPETSPESV⁴⁵⁰) and (iv) the C-terminus (⁵⁴⁷GNKLFACGPDAKEVRKENQANTSVV⁵⁷¹).

An alignment of functionally characterized ferroportin homologs from *A. fumigatus*, *Arabidopsis thaliana*, *B. bacteriovorus*, zebrafish (*Danio rerio*) and *Homo sapiens* is shown in Fig. 6[26,30,31,36]. The major differences in the protein domains in *A. fumigatus* FpnA compared to human Fpn1 are two truncations, one of 16 amino acid residues in the cytosolic loop between TM6 and TM7 and the other one of 52 amino acid residues in the extracellular (corresponding to vacuolar lumen in FpnA) loop between TM9 and TM10. *A. fumigatus* FpnA shares the truncation in the loop between TM9 and TM10 with the plant and bacterial ferroportin homologs but not with that of zebrafish, indicating that the insertion is a metazoan feature. The 16 amino acid residue-truncation in the loop between TM6 and TM7 is also found in the bacterial version, whereas the plant and zebrafish homologs have shorter truncations.

## Discussion

In certain ecosystems, microorganisms are exposed to toxic metal concentrations, either naturally or as a result of increasing industrialization. Four mechanisms of metal toxicity have been proposed[37]: (i) substitution of essential metals of metalloproteins, (ii) binding to catalytic residues of non-metalloenzymes, (iii) allosteric inhibition of enzymes by binding outside of the catalytic site and (iv) induction of oxidative stress. In turn, microorganisms developed resistance mechanisms to cope with metal toxicity. The resistance mechanisms are categorized into biosorption, biotransformation, compartmentalization, complexation, and cellular efflux strategies[38]. Several mechanisms mediating metal resistance of *A. fumigatus* have been elucidated at the molecular level. This mold species detoxifies excess iron by vacuolar deposition mediated by the transporter CccA[13]. Similarly, excess zinc is detoxified by vacuolar compartmentalization mediated by the transporter ZrcA[39,40]. Copper is detoxified by cellular export mediated by the P(1B)-type ATPase CrpA[41]. In contrast, several fungal species including *Cryptococcus neoformans* employ copper complexation by cysteine-rich proteins, termed metallothioneins[42], suggesting that resistance mechanisms can be species-specific even within the fungal kingdom. The copper exporter CrpA is also involved in zinc detoxification[43], demonstrating that resistance mechanisms can act metal-overlapping and show redundancy as zinc is also detoxified by vacuolar compartmentalization as described above. For cadmium detoxification, *A. fumigatus* uses the P(1B)-type ATPase PcaA[40]. Arsenic detoxification is mediated by methylation and cellular export involving the methyltransferase ArsM and the efflux metalloid/H+ antiporter AcrA[44,45]. Interestingly, the major facilitator-type efflux transporter for catabolic purine intermediates and toxic purine analogues, termed NmeA, is also crucial for resistance to cadmium, zinc and borate in *Aspergillus nidulans*[46]. In contrast, resistance mechanisms to nickel, cobalt and gallium were unknown so far. With significant species specificity, low levels of some potentially toxic metals are required for certain

biological processes. For example, cobalt is essential for biosynthesis of cobalamins such as vitamin B12, which, however, appears to be dispensable for most fungal species[47]. For *A. fumigatus*, metals with biological functions include iron, copper, zinc, and nickel, as opposed to, e.g., cobalt, gallium, aluminum and cadmium.

Here we demonstrate that *A. fumigatus* utilizes a homolog of human Fpn1, termed FpnA, for metal homeostasis. Notably, the exon-intron structure of the gene was partially incorrectly predicted in several fungal genomes, including that of *A. fumigatus*. After correction, *A. fumigatus* FpnA and human Fpn1 show an identity of 25% and represent the best homologs in reverse blastp searches with an E-value of 3e⁻⁴⁷ for FpnA versus the human proteome. Inactivation of FpnA increased susceptibility of *A. fumigatus* mainly to nickel but also to cobalt and gallium, while artificial transcriptional up-regulation increased the resistance to these metals compared to WT. Localization of functionally Venus-tagged FpnA in the vacuolar membrane suggests that FpnA most likely detoxifies nickel, cobalt and gallium by vacuolar compartmentalization in *A. fumigatus*. Previously, zinc and iron have also been shown to be detoxified by vacuolar deposition[13,39]. Consistent with compartmentalization of nickel, overproduction of FpnA reduced the utilization of urea as the sole nitrogen source, most likely by depriving cytosolic urease of its essential cofactor nickel[27]. In line, *fpnA* overexpression reduced alkalinization with urea as nitrogen source, indicating impaired urease activity. The fact that overexpression of *fpnA* did not cause growth defects on standard minimal medium with glutamine as nitrogen source (Fig. 1), suggests that FpnA does not transport metals, other than nickel, that are essential for cellular functions. Unlike transporters mediating detoxification of copper, zinc, cadmium, aluminum or iron[13,39–41,43,45], *fpnA* transcript levels did not respond to short-term exposure to the identified metal substrates. Thus, the role of FpnA in metal resistance could not have been identified by commonly used transcriptome response analyses.

The transition metals nickel and cobalt are neighbors of iron in the periodic table and therefore share some properties with iron. The most common oxidation states of cobalt are (+2) and (+3) and that of nickel is (+2). In addition to anthropogenic contamination, these metals can leach from naturally occurring minerals into soil and freshwater habitats to reach toxic concentrations. Noteworthy, the toxicity of cobalt is largely related to its interference with iron homeostasis[48]. Gallium, a rather rare post-transition metal found primarily in the (+3) oxidation state, has an ionic radius almost identical to that of iron, can be taken up by cellular iron transport systems and can replace iron in iron-containing proteins[49]. Unlike iron, gallium cannot be reduced under physiological conditions[50], which inhibits the functionality of gallium-complexed proteins and arrests cell growth[49]. Gallium-based compounds show potential as antibacterial agents[51] and have been reported to have antifungal activity[52]. In addition, gallium compounds have several medical applications including its use as a diagnostic and therapeutic agent in cancer and disorders of calcium and bone metabolism[53]. Recently, the radionuclide gallium-68 complexed with siderophores has shown promising potential for targeted imaging of bacterial and fungal infections using positron emission tomography (PET)[54]. Taken together, these data illustrate that nickel, cobalt and gallium share properties similar to those of iron.

In contrast to nickel, cobalt and gallium, neither inactivation nor overexpression of FpnA significantly affected growth at toxic concentrations of iron, copper, zinc, cadmium or aluminum. Even in a genetic background with increased iron susceptibility and apparent saturation of CccA-mediated iron detoxification due to inactivation of the iron uptake repressor SreA[13], inactivation or overexpression of *fpnA* did not affect growth in the presence of toxic iron concentrations. These data suggest that FpnA is not involved in the detoxification of iron, copper, zinc, cadmium and aluminum. Notably, specific resistance mechanisms for the latter metals have been previously identified in *A. fumigatus* (see above), which may explain the metal substrate exclusivity found here. Functional studies in *Xenopus* oocytes demonstrated that human Fpn1 transports metals other than iron, such as cobalt, zinc, and nickel, but not copper or cadmium[23,24]. Therefore,

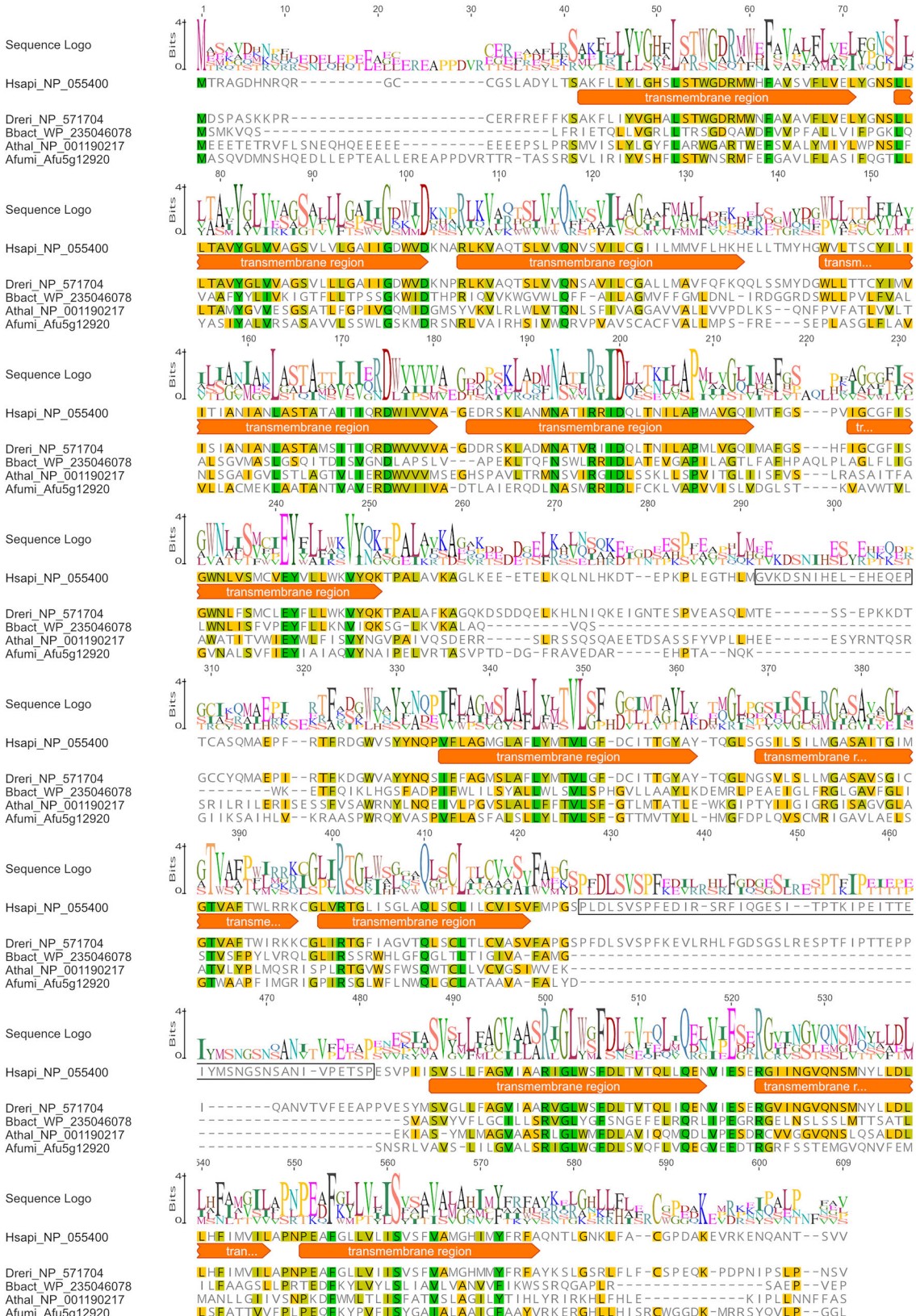

**Fig. 6 | Amino acid alignment of ferroportin homologs from *Homo sapiens* (Hsapi), *Danio rerio* (Dreri), *Bdellovibrio bacteriovorus* (Bbact), *Arabidopsis thaliana* (Athal), *Aspergillus fumigatus* (Afumi).** Human Fpn1 regions framed in black are missing in *A. fumigatus* FpnA and other homologs (see text). The differences are also illustrated in the structural alignments shown in Supplementary Fig. S2. A MUSCLE alignment of the protein sequences was performed using Geneious Prime. The annotations of the TMs were obtained from the deposited sequence (NP_055400) of NCBI database[64]. The similarities of amino acids are indicated using the score matrix Blosum62: green corresponds to 100%, gold corresponds to 80–99% and yellow corresponds to 60-79% amino acid similarity.

*A. fumigatus* FpnA and human Fpn1 share the common substrates nickel and cobalt, but not iron, which is the physiologically relevant substrate of human Fpn1.

Even though the extensively studied fungal species *S. cerevisiae, C. albicans* and *S. pombe* lack FpnA homologs, FpnA was found to be conserved in all divisions of the fungal kingdom and several eukaryotic lineages apart from mammals, which indicates an important role in diverse habitats. A bacterial Fpn1 homolog from *B. bacteriovorus*, with elucidated crystal structure, was shown to mediate in vitro transport of iron cobalt, nickel and manganese when reconstituted into liposomes[31]. Fpn1 homologs from plants, termed IREG, have been reported to mediate cellular export as well as vacuolar storage of nickel in nickel-hyperaccumulating plant species[28,29]. Moreover, two Fpn1 homologs of *Arabidopsis thaliana* localized in the plasma membrane and the vacuolar membrane, respectively, have been shown to function in iron and cobalt homeostasis[30]. Phylogenetically, animals and fungi are sister groups while plants constitute an independent evolutionary lineage, i.e., animals and fungi are each other's closest relatives[55]. Therefore, the divergent localization and functionalization of ferroportin homologs from a common ancestor in these two sister groups is of particular evolutionary interest.

Human Fpn1 displays a MFS-type fold[26] and structural prediction suggests a highly similar structure of *A. fumigatus* FpnA. Remarkably, the ferroportin homologs of *A. fumigatus*, *A. thaliana* and *B. bacteriovorus* display a truncation of about 52 amino acid residues in the loop between TM9 and TM10 to those of the compared human and fish homologues. The metazoen Fpn1 homologs are the only known cellular iron exporters and are subject to post-translational regulation by the hormone hepcidin, which binds and signals the endocytosis and proteolysis of the ferroportin-hepcidin complex when systemic iron levels are high[56]. The expected absence of hepcidin regulation in the non-metazoan ferroportin homologs including that of *A. fumigatus* is a likely explanation for some of the protein differences described.

The fungal resource has great potential for bioremediation of metal contaminated sites, metal nanoparticle synthesis and biomining[38]. Increasing metal resistance is an important tool for improving such biotechnological strategies. As shown here, transcriptional up-regulation of FpnA significantly increased resistance to nickel, cobalt and gallium and therefore FpnA may represent a valuable target in this regard.

## Methods
### Growth conditions
For spore production and growth assays, *A. fumigatus* strains were grown on solid or liquid *Aspergillus* minimal medium[57] containing 1% (w/v) glucose and 20 mM glutamine as carbon and nitrogen sources, respectively, with the addition of 0.01 mM $FeSO_4$ as iron source. For phenotyping on solid media, $10^4$ conidia were spot-inoculated and plates were incubated for 37 °C for 48 h. For liquid shake-flask cultures, 100 mL minimal medium in 500 mL flasks were inoculated with $10^6$ spores per mL of medium and incubated at 37 °C shaking at 200 rpm. Supplementation with xylose was used to induce P*xylP* promoter-mediated gene expression and is indicated in the respective figures[19,20]. The metals tested are indicated in the respective figures: $CuSO_4$, $ZnCl_2$ (Merck); $FeSO_4$ (Scharlau); $CoSO_4$ (Serva); $Ga(NO_3)_3$ (Sigma-Aldrich); $Al_2(SO_4)_3$, $CdSO_4$, $NiCl_2$ (Thermo Scientific).

### Generation of *A. fumigatus* mutant strains
All studies were carried out using *A. fumigatus* strain AfS77 (termed WT here), a derivative of *A. fumigatus* ATCC46645 lacking non-homologous recombination (Δ*akuA::loxP*) to facilitate genetic manipulation[58,59]. To functionally characterize FpnA (Afu5g12920), the encoding *fpnA* gene was deleted by replacement with the hygromycin resistance gene (*hph*), resulting in the strain Δ*fpnA*. Moreover, *fpnA* was conditionally expressed under control of the *xylP* promoter (P*xylP*) with (*fpnA*[PxylP-Venus] strain) or without (*fpnA*[PxylP] strain) N-terminal tagging with the yellow fluorescent protein derivative Venus[19–21].

For the generation of the Δ*fpnA* strain, *fpnA* flanking 5′ and 3′ non-coding regions (NCR) were amplified from WT gDNA using primers *OH003_5'fpnA_fwd/OH004_5'fpnA_rev* and *OH007_3' fpnA_fwd/OH008_3'fpnA_rev*, respectively (a summary of all primers is listed in Supplementary Table S2). The *hph* resistance cassette was amplified from plasmid pAN7-1 using primers *OH005_hph_fwd/OH006_hph_rev*[60]. The pUC19 plasmid backbone (Thermo Fisher), was amplified with primers *OH001_pUC19_fwd/OH002_pUC19_rev*. Amplified fragments were ligated using the NEBuilder© HiFi DNA Assembly (New England Biolabs). For fungal transformation, the Δ*fpnA* transformation cassette (4.5 kb) was PCR-amplified from pOH001 using primers *OH009_TCA_fwd/OH010_TCA_rev*. Successfully transformed colonies were selected on AMM with 0.1 mg/mL hygromycin (Calbiochem).

For conditional *fpnA* expression in strain *fpnA*[PxylP] flanking 5′ NCR and *fpnA* coding sequence (cds) were amplified from WT gDNA using primers *OH003_5'fpnA_fwd/OH011_5'fpnA_rev* and *OH016_fpnA_cds/OH017_fpnA_cds_rev*, respectively. The *hph* resistance cassette was amplified from plasmid pAN7-1[60] using primers *OH012_hph_fwd/OH013_hph_rev*. For conditional *fpnA* expression, P*xylP* was applied, which was amplified from plasmid pSO16 using primers *OH014_PxylP_fwd/OH015_PxylP_rev*[19,20,61]. The pUC19 plasmid backbone (Thermo Fisher), was amplified with primers *OH001_pUC19_fwd/OH002_pUC19_rev*. The fragments were assembled using the NEBuilder© HiFi DNA Assembly (New England Biolabs), yielding plasmid pOH002. The TC (5.2 kb) used for fungal transformation was amplified by PCR using primers *OH018_TCA_fwd/OH019_TCA_rev*. For generation of the N-terminally Venus-tagged FpnA strain (*fpnA*[PxylP-Venus] strain) the same approach as described for construction of plasmid pOH002 was applied using the following primer pairs: *OH001_pUC19_fwd/OH002_pUC19_rev*, *OH003_5'fpnA_fwd/OH011_5'fpnA_rev*, *OH023_fpnA_cds_fwd/OH017_fpnA_cds_rev*, *OH012_hph_fwd/OH013_hph_rev* and *OH014_PxylP_fwd/OH020_PxylP_rev*. The Venus for N-terminal tagging was amplified from plasmid pSO16[61] using primers *OH021_venus_fwd/OH022_venus_rev*. Respective TC (5.9 kb) was amplified using primers *OH018_TCA_fwd/OH019_TCA_rev*. The selection of transformants was also done using hygromycin.

Genetic transformation of *A. fumigatus* AfS77 was performed according to Tilburn et al. [62] and confirmed by Southern blot analysis. Schematic depiction of the gene manipulations is shown in Supplementary Fig. S5. All *A. fumigatus* strains generated and used in this study are listed in Supplementary Table S3.

### RNA isolation and Northern blot analysis
Total RNA was isolated from mycelia harvested from liquid cultures using TRI Reagent (Sigma-Aldrich) according to the manufacturer's protocol. 10 μg of total RNA were separated on a 1.2% agarose gel with 1.85% (w/v) formaldehyde. The gels were blotted onto a Hybond™-N⁺ membrane (Amersham Biosciences). PCR-amplified, digoxigenin-labelled probes were used for gene-specific hybridization. Primers used for the generation of the Northern blot hybridization probes are listed in Supplementary Table S2.

### Fluorescence microscopy
Fungal cultures for fluorescence microscopy were prepared in 8-well chamber slides (μ-Slide 8 Well; Ibidi) with $10^4$ spores per well in a total volume of 200 μL minimal medium without iron and 0.05 μM $NiCl_2$. Cultures were incubated for 15 h at 37 °C, followed by an 1 h incubation under supplementation with 0.1% xylose (for induction of Venus-tagged FpnA expression) and CellTracker™ Blue CMAC Dye (Invitrogen™) in a final concentration of 1 μM for visualization of the vacuolar lumen[25]. Fluorescence microscopy was performed using an inverted Eclipse Ti2-E microscope (Nikon) with a 60× TIRF objective (Plan-APOCHROMAT 60×/1.49 Oil, Nikon) combined with a spinning disk confocal unit (CSU-W1, Yokogawa), an EMCCD camera (iXon Ultra 888, Andor) and an additional 1.5× magnification. Microscopy pictures were analyzed and deconvolved using the NIS-Elements software (Nikon), applying the Lucy-

Richardson algorithm. Further processing of microscopy pictures was done using Fiji ImageJ[63].

## Bioinformatics

For designing the transformation constructs, Benchling (Biology Software; 2024; https://benchling.com) was used. The protein sequences of *A. fumigatus* FpnA, accession number Afu5g12920-T, was obtained from FungiDB (fungidb.org) and Apollo/VEuPathDB (apollo.veupathdb.org), respectively. Proteins were blasted using Geneious Prime 2024.0.5 (https://www.geneious.com), which accesses the NCBI database[64]. Alignments and phylogenetic tree were made using Geneious Prime. The protein structure of human ferroportin, Fpn1 (PDB DOI: 6WBV; https://doi.org/10.2210/pdb6wbv/pdb) was obtained from the Protein Data Bank (rcsb.org)[32]. In addition, the predicted structure was obtained from AlphaFold (https://alphafold.ebi.ac.uk/)[35]. The protein structure of *A. fumigatus* FpnA (Afu5g12920-T), was predicted using ColabFold (AlphaFold2 using MMseqs2)[34]. The DALI Z-score was calculated on the website http://ekhidna2.biocenter.helsinki.fi/dali/[33]. Visualization and structural alignments of the proteins were generated using PyMOL Molecular Graphics System, Version 3.0.3, Schrödinger, LLC.

## Statistics and Reproducibility

Figures show exemplary results from three biological replications. All replication attempts were successful.

## Reporting summary

Further information on research design is available in the Nature Portfolio Reporting Summary linked to this article.

## Data availability

The authors declare that the data supporting the findings of this study are available within the article and the Supplementary Information file, or from the corresponding author upon request. Uncropped blot images are shown in the Supplementary Fig. S6.

## Abbreviations

| | |
|---|---|
| CDS | coding sequence |
| EM | electron microscopy |
| MFS | major facilitator superfamily |
| NCR | non-coding region |
| PDB | protein data base |
| TC | transformation cassette |
| TM | transmembrane region |
| WT | wild type |

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

## Acknowledgements

We thank Mika Jormakka for inspiring discussions and valuable input. This research was funded by the Austrian Science Fund (FWF) [Grant-DOI: 10.55776/DOC82 and 10.55776/I6613]. For open access purposes, the author has applied a CC BY public copyright license to any author accepted manuscript version arising from this submission. The authors acknowledge support from COST Action (European Cooperation in Science and Technology), FeSImmChemNet, CA21115. J.P. was supported by a fellowship of the Higher Education Commission (HEC) of Pakistan.

## Author contributions

I.H.: conceptualization, methodology, investigation, visualization, supervision, writing—original draft, writing—review & editing, software; S.O.: conceptualization, methodology, investigation, visualization, supervision, writing—original draft, writing—review & editing, software; B.A.: methodology, investigation; A.Y.: conceptualization, methodology, supervision; P.C.: methodology, investigation; M.A.: methodology, investigation; J.P.: methodology, investigation; G.G.: methodology, investigation; M.M.: conceptualization, methodology, investigation, supervision; C.D.: conceptualization, funding, acquisition, project

administration, supervision, writing—review & editing, resources; H.H.: conceptualization, funding acquisition, project administration, supervision, writing—original draft, writing—review & editing, resources. All authors have read and approved the final manuscript.

## Competing interests

The authors declare no competing interests.
