## [Transparent Peer Review file · Communications Biology]

FpnA, the *Aspergillus fumigatus* homolog of human ferroportin, mediates resistance to nickel, cobalt and gallium but does not function in iron homeostasis

Corresponding Author: Professor Hubertus Haas

Version 0:

Reviewer comments:

Reviewer #1

(Remarks to the Author)

In this study, authors demonstrated functions of a putative homolog of the vertebrate iron transporter ferroportin (Fpn1), termed FpnA, at last, they found FpnA is not involved in regulation for iron homeostasis but mediates resistance to nickel, cobalt and gallium by a series of strategies including gene deletion and overexpression. Findings are very interesting and unpredicted important. Writing is very clear and logic, data are able support conclusions.

Some of revised suggestions:

1. Title is not very revised and could not cover all data and main findings in this study : it should include FpnA, in addition, metal includes iron?
2. Based on colony phenotypes, deletion of FpnA showed hypersensitive to nickel and and OE showed a contrast result such that they concluded that FpnA mediates resistance to nickel, cobalt and gallium but not to iron, cadmium, aluminum or zinc, how much high concentration could be referred as resistance to metal?
3. Based on co-localization of FpnA with vacuolar marker, they concluded that FpnA detoxifies substrate metals by vacuolar deposition, metals should be overloaded in vacuoles, it seems like not very solid, authors should discuss it.
4. In Fig. 4, overexpression of fpnA (fnpAPxyIP and fnpAPxyIP-Venus strains with xylose) impairs utilization of urea as sole nitrogen source supplementation with 0.5 mM NiCl₂. What is rational for determining the concentration of NiCl₂
5. Fig. 6 could be moved to Suppl materials since readers are unable to obtain conclusions from these data.

Reviewer #2

(Remarks to the Author)

This is an important and interesting paper on the putative physiological functions of *A. fumigatus* FpnA. The study is well-planned and well-performed and I can suggest only some minor points to be considered.

- 1) It would be worth seeing expression changes for the urease gene in the fpnA gene deletion and overexpression strains under normal and Ni-supplemented culture conditions. The hypothesized relationship between urease and FpnA as a Ni-transporter seems to be still indirect and conditional, which needs further verification.
- 2) To confirm highly conserved functions for FpnA in the kingdom Fungi, expression of fpnA in baker's yeast as a clean box could be recommended. Most likely, simplification of the genome of *S. cerevisiae* led to the evolutionary loss of fpnA ortholog.

I would recommend Point 1) to consider to strengthen the manuscript further and please regard Point 2) as optional.

A minor point:

"In contrast to" seems to be one of the authors' favorite phrases throughout the manuscript. Please replace with synonyms wherever possible.

Reviewer #3

(Remarks to the Author)

The authors report the identification and functional characterization of a ferroportin homolog in the human fungal pathogen *Aspergillus fumigatus*. As stated in the manuscript, this is the first study to suggest that most Ascomycota and Basidiomycota possess a ferroportin homolog, although a few model fungi, such as *S. cerevisiae* and *S. pombe*, do not. The data indicate that FpnA in *A. fumigatus* mediates resistance to nickel, cobalt, and gallium, but not to iron, and that the protein localizes to the vacuolar membrane. Notably, the reduced utilization of urea as a nitrogen source caused by the overproduction of FpnA strongly supports the protein's function. I also appreciate the authors' effort to re-annotate the exon-intron sequence of the gene.

Overall, I found the manuscript to be well-organized, clearly written, and it presents the data effectively. It is worthy of publication.

Version 1:

Reviewer comments:

Reviewer #1

(Remarks to the Author)

I think authors have revised thoroughly and response my comments point by point. They changed Title and revised the manuscript carefully. I have no more comments.

Reviewer #2

(Remarks to the Author)

The criticisms raised by me were addressed adequately. I suggest the acceptance of this contribution in its present form.

Point-by-point response

Reviewer #1 (Remarks to the Author):

In this study, authors demonstrated functions of a putative homolog of the vertebrate iron transporter ferroportin (Fpn1), termed FpnA, at last, they found FpnA is not involved in regulation for iron homeostasis but mediates resistance to nickel, cobalt and gallium by a series of strategies including gene deletion and overexpression. Findings are very interesting and unpredicted important. Writing is very clear and logic, data are able support conclusions.

Some of revised suggestions:

1. Title is not very prevised and could not cover all data and main findings in this study : it should include FpnA, in addition, metal includes iron?

The title was changed to "FpnA, the *Aspergillus fumigatus* homolog of human ferroportin, mediates resistance to nickel, cobalt and gallium but does not function in iron homeostasis"

2. Based on colony phenotypes, deletion of FpnA showed hypersensitive to nickel and OE showed a contrast result such that they concluded that FpnA mediates resistance to nickel, cobalt and gallium but not to iron, cadmium, aluminum or zinc, how much high concentration could be referred as resistance to metal?

As commonly used in the literature (e.g., reference 37), we do not use the term "resistance" as a definition of a certain concentration (as used for clinical resistance) but relative to the wild type reference strain. This is possible as we work with isogenic mutants differing only in a single gene or allele, i.e. "decreased resistance" means decreased tolerance compared to the reference wild type strain. Notably, our terminology was not criticized by the other referees.

3. Based on co-localization of FpnA with vacuolar marker, they concluded that FpnA detoxifies substrate metals by vacuolar deposition, metals should be overloaded in vacuoles, it seems like not very solid, authors should discuss it.

We concluded this in analogy to other metal resistance mechanisms, e.g. Ccc1/CccA-mediated vacuolar iron deposition and Zrc1/ZrcA-mediated vacuolar zinc deposition of iron detoxification, which are conserved in *A. fumigatus* and *S. cerevisiae* and are mentioned in lines 222-225. We changed the wording in line 138 to "... these results suggest that FpnA most likely detoxifies the substrate metals by transport from the cytoplasm into the vacuolar lumen." and at lines 247-249 to "Localization of functionally Venus-tagged FpnA in the vacuolar membrane suggests that FpnA most likely detoxifies nickel, cobalt and gallium by vacuolar compartmentalization in *A. fumigatus*."

4. In Fig. 4, overexpression of fpnA (fpnAPxyIP and fpnAPxyIP-Venus strains with xylose) impairs utilization of urea as sole nitrogen source supplementation with 0.5 mM NiCl₂. What is rational for determining the concentration of NiCl₂

The goal of this experiment was to elucidate whether overexpression of *fpnA* increases the nickel resistance of the *A. fumigatus* wild type (WT) strain used. Therefore, we used a nickel concentration that allows only minimal growth of the used *A. fumigatus* WT strain and compared it with the mutant strains (*fpnA* overexpression and deletion) generated in the same genetic background.

5. Fig. 6 could be moved to Suppl materials since readers are unable to obtain conclusions from these data.

Fig. 6 shows the amino acid residues conserved in Fpn1/FpnA homologs and the regions missing in *A. fumigatus* FpnA compared to human Fpn1. We therefore believe that this figure is informative for

the reader. We corrected the following sentence of the legend: "Human Fpn1 regions framed in black are missing in *A. fumigatus* FpnA and other homologs (see text)."

We thank the Reviewer for his comments and suggestions to improve the study and the presentation.

Reviewer #2 (Remarks to the Author):

This is an important and interesting paper on the putative physiological functions of *A. fumigatus* FpnA. The study is well-planned and well-performed and I can suggest only some minor points to be considered.

1) It would be worth seeing expression changes for the urease gene in the *fpnA* gene deletion and overexpression strains under normal and Ni-supplemented culture conditions. The hypothesized relationship between urease and FpnA as a Ni-transporter seems to be still indirect and conditional, which needs further verification.

As suggested, we analyzed the expression of the *A. fumigatus* urease encoding gene, termed *ureB* (reference 27), by Northern blot analysis. Interestingly, nickel supplementation and *fpnA* deletion, but not *fpnA* overexpression, decreased *ureB* transcript levels, possibly indicating negative regulation of *ureB* expression by nickel. The link between *ureB* transcript levels and nickel could be indirect through the positive effect of nickel on urease activity (see below), i.e., increased urease activity increases urea utilization, which could downregulate *ureB* expression. The Northern blot analysis is shown in the new Supplementary Figure S2; the data are discussed in the main text at lines 153-159.

To further investigate the growth defect on urea caused by *fpnA* overexpression, we analyzed urease activity. Urease activity, i.e., hydrolysis of urea, leads to the formation of ammonia and hence alkalization, which can be detected by the pH-sensitive color change of phenol red (PMID: 16561200 and reference 27). As shown in the new Supplementary Fig. S3, nickel supplementation enhanced reddish coloration of mycelia of WT, $\Delta fpnA$ and *fpnA*^{P_{xyIP}} strains, which indicates increased urease activity. These results suggest that nickel is a limiting factor for urease activity in the growth medium used. Compared to WT and $\Delta fpnA$ strains, the *fpnA*^{P_{xyIP}} mutant displayed reduced reddish coloration of its mycelia without, but not with, nickel supplementation, supporting the hypothesis that *fpnA* overexpression without nickel supplementation reduces urease activity, most likely due to vacuolar nickel deposition. These results are discussed in the main text at lines 159-166 and 252-253.

2) To confirm highly conserved functions for FpnA in the kingdom Fungi, expression of *fpnA* in baker's yeast as a clean box could be recommended. Most likely, simplification of the genome of *S. cerevisiae* led to the evolutionary loss of *fpnA* ortholog.

I would recommend Point 1) to consider to strengthen the manuscript further and please regard Point 2) as optional.

We agree that it would be interesting to functionally characterize *A. fumigatus fpnA* by heterologous expression in *S. cerevisiae*. As a matter of fact, heterologous expression of plant *fpnA* homologs have been shown to mediate nickel resistance (PMID: 38527955). However, heterologous expression always bears the risk of incompatibilities and artefacts. As we were able to functionally characterize FpnA in the original organism, we did not initiate its characterization in yeast.

A minor point:

"In contrast to" seems to be one of the authors' favorite phrases throughout the manuscript. Please replace with synonyms wherever possible.

We reworded several "in contrast to", e.g. line 28, line 56, line 76, line 126, line 142, line 239, and line 255.

We thank the Reviewer for his comments and suggestions to improve the study and the presentation.

Reviewer #3 (Remarks to the Author):

The authors report the identification and functional characterization of a ferroportin homolog in the human fungal pathogen *Aspergillus fumigatus*. As stated in the manuscript, this is the first study to suggest that most Ascomycota and Basidiomycota possess a ferroportin homolog, although a few model fungi, such as *S. cerevisiae* and *S. pombe*, do not. The data indicate that FpnA in *A. fumigatus* mediates resistance to nickel, cobalt, and gallium, but not to iron, and that the protein localizes to the vacuolar membrane. Notably, the reduced utilization of urea as a nitrogen source caused by the overproduction of FpnA strongly supports the protein's function. I also appreciate the authors' effort to re-annotate the exon-intron sequence of the gene.

Overall, I found the manuscript to be well-organized, clearly written, and it presents the data effectively. It is worthy of publication.

We thank the Reviewer for his supporting summary!